# TRRAP Enhances Cancer Stem Cell Characteristics by Regulating NANOG Protein Stability in Colon Cancer Cells

**DOI:** 10.3390/ijms24076260

**Published:** 2023-03-26

**Authors:** Kyung-Taek Kang, Min-Joo Shin, Hye-Ji Moon, Kyung-Un Choi, Dong-Soo Suh, Jae-Ho Kim

**Affiliations:** 1Department of Physiology, College of Medicine, Pusan National University, Yangsan 50612, Gyeongsangnam-do, Republic of Korea; 2Department of Pathology, College of Medicine, Pusan National University, Yangsan 50612, Gyeongsangnam-do, Republic of Korea; 3Department of Obstetrics and Gynecology, School of Medicine, Pusan National University, Yangsan 50612, Gyeongsangnam-do, Republic of Korea; 4Convergence Stem Cell Research Center, Pusan National University, Yangsan 50612, Gyeongsangnam-do, Republic of Korea

**Keywords:** cancer stem cells, TRRAP, NANOG, ubiquitination, protein stability, proteasome

## Abstract

NANOG, a stemness-associated transcription factor, is highly expressed in many cancers and plays a critical role in regulating tumorigenicity. Transformation/transcription domain-associated protein (TRRAP) has been reported to stimulate the tumorigenic potential of cancer cells and induce the gene transcription of NANOG. This study aimed to investigate the role of the TRRAP-NANOG signaling pathway in the tumorigenicity of cancer stem cells. We found that TRRAP overexpression specifically increases NANOG protein stability by interfering with NANOG ubiquitination mediated by FBXW8, an E3 ubiquitin ligase. Mapping of NANOG-binding sites using deletion mutants of TRRAP revealed that a domain of TRRAP (amino acids 1898–2400) is responsible for binding to NANOG and that the overexpression of this TRRAP domain abrogated the FBXW8-mediated ubiquitination of NANOG. TRRAP knockdown decreased the expression of CD44, a cancer stem cell marker, and increased the expression of P53, a tumor suppressor gene, in HCT-15 colon cancer cells. TRRAP depletion attenuated spheroid-forming ability and cisplatin resistance in HCT-15 cells, which could be rescued by NANOG overexpression. Furthermore, TRRAP knockdown significantly reduced tumor growth in a murine xenograft transplantation model, which could be reversed by NANOG overexpression. Together, these results suggest that TRRAP plays a pivotal role in the regulation of the tumorigenic potential of colon cancer cells by modulating NANOG protein stability.

## 1. Introduction

Colorectal cancer is a malignancy of the inner wall of the large intestine [1] and the third most common cause of cancer-related deaths in both men and women worldwide [2]. Several diagnostic markers of colon cancer associated with recurrence or metastases have been identified and are now being adopted in clinical applications [3]. Cancer stem cells (CSCs) are a subpopulation of cancer cells that can self-renew and have high tumorigenic potential. CSCs are resistant to conventional drug therapy and have considerable metastatic potential and tumorigenic activity [4]. Recent studies have highlighted the involvement of colon CSCs in cancer recurrence. Colon CSCs have a significant tumorigenic niche due to their overexpression of self-renewal-associated genes [5]. Thus, identifying new diagnostic and therapeutic targets for CSCs is required to treat colorectal cancer.

NANOG, a homeobox protein, is a transcription factor that contributes to embryonic stem cell pluripotency and self-renewal [6]. It has been reported that NANOG is a key regulator and transcription factor of CSCs and is upregulated in various human cancers [7]. NANOG has been reported to act as a tumor marker in patients with colorectal cancer and is associated with clinical and pathological features [8]. In human colorectal cancer, NANOG modulates stemness and tumorigenicity [9]. A high expression of NANOG in HCT116 cells causes increased colony formation and tumorigenicity by controlling the CSC population [10]. RNAi-mediated silencing of NANOG expression suppressed proliferation and increased apoptosis in the EpCAM^+^/CD44^+^ CSC population of HCT116 colorectal cancer cells [11]. In addition to transcriptional regulation, NANOG protein levels are regulated by ubiquitination and proteasomal degradation [12,13]. FBXW8 is a well-known ubiquitin ligase that regulates NANOG protein stability by mediating its ubiquitination [14]. However, the molecular mechanisms underlying the regulation of NANOG protein stability have not yet been elucidated.

TRRAP is a large adaptor protein with homology similar to PIKK kinases [15]. TRRAP promotes histone acetylation and chromatin remodeling and regulates gene expression and embryonic development [16]. TRRAP depletion has been shown to decrease the expression of the stemness-associated genes *OCT4*, *SOX2*, and *NANOG* and increase the expression of differentiation markers from the germ layers [16]. TRRAP expression levels have been reported to be significantly upregulated in breast cancer [17], while the TRRAP knockdown reduced the CSC-like properties of glioma [18,19]. TRRAP overexpression also increased the mRNA levels of *NANOG*, while TRRAP knockdown reduced tumor growth in a murine ovarian cancer xenograft model [20]. However, although TRRAP and NANOG play an important role in the tumorigenicity of CSCs, the role of TRRAP in regulating NANOG protein stability remains uncertain.

In this study, we found that TRRAP regulates the protein stability of NANOG in CSCs. Moreover, the molecular mechanisms associated with the TRRAP-dependent regulation of the ubiquitination and proteasomal degradation of NANOG were investigated to clarify the role of TRRAP in the tumorigenicity of colorectal CSCs.

## 2. Results

### 2.1. TRRAP Increases NANOG Protein Levels by Interfering with the Proteasomal Degradation of NANOG

TRRAP overexpression has been reported to increase the mRNA levels of endogenous NANOG [20]. Notably, we observed that TRRAP overexpression increased the mRNA and protein levels of NANOG when FLAG-TRRAP and TAP-NANOG were co-transfected in HEK293 cells (Figure 1A), suggesting that FLAG-TRRAP overexpression may increase the protein stability of TAP-NANOG. To explore whether TRRAP interacts with NANOG, co-immunoprecipitation experiments with TRRAP and NANOG were performed. After co-transfection of FLAG-TRRAP and TAP-NANOG in HEK293 cells, FLAG-TRRAP was immunoprecipitated with an anti-FLAG antibody. As shown in Figure 1A, TAP-NANOG was co-precipitated with FLAG-TRRAP in HEK293 cells. To explore whether NANOG can directly interact with TRRAP, TAP-NANOG was purified via affinity purification with streptavidin agarose and eluted using biotin, followed by incubation with the immunoprecipitants of FLAG-TRRAP. TAP-NANOG could be pulled down by FLAG-TRRAP (Appendix A), suggesting the direct interaction between TAP-NANOG and FLAG-TRRAP.

To explore whether NANOG protein levels are specifically increased by TRRAP overexpression, we overexpressed OCT4, SOX2, or NANOG along with TRRAP. In contrast to the TRRAP-dependent increase in NANOG protein levels, there was no significant change in OCT4 and SOX2 protein levels (Figure 1B). Treatment of HEK293 cells with the proteasome inhibitor MG132 to block proteasomal degradation of proteins increased the protein levels of NANOG and OCT4. However, the mRNA levels of *OCT4*, *SOX2*, and *NANOG* were not significantly affected by TRRAP overexpression (Figure 1C). These results suggest that TRRAP specifically increased the protein level of NANOG, but not OCT4 or SOX2, suggesting the TRRAP-dependent regulation of NANOG protein stability.

To explore the role of TRRAP in the protein stability of NANOG, HEK293 cells were transfected with TAP-NANOG along with FLAG-TRRAP, followed by treatment with CHX, a protein synthesis inhibitor. In the control group, the protein levels of NANOG decreased in a time-dependent manner; however, the overexpression of TRRAP increased the protein levels of NANOG and inhibited its degradation (Figure 1D), suggesting that TRRAP protein increases the protein stability of NANOG. To clarify whether the stability of NANOG is regulated by the ubiquitin-proteasome system, we performed a ubiquitination assay. We observed that NANOG ubiquitination levels decreased in response to TRRAP overexpression (Figure 1E). These data indicate that TRRAP stabilizes NANOG by suppressing its ubiquitination and proteasomal degradation.

### 2.2. Identification of the NANOG-Binding Domain of TRRAP

Since TRRAP is a protein with a high molecular weight (~430 kDa), we next characterized the domains of TRRAP involved in its interaction with NANOG. Five deletion mutants of TRRAP were produced to map the domains of TRRAP that interact with NANOG (Figure 2A). FLAG-tagged full-length TRRAP and five deletion mutants of TRRAP were co-transfected with TAP-NANOG in HEK293 cells, and the cell lysates were subjected to immunoprecipitation with anti-FLAG antibodies. Western blotting demonstrated that NANOG co-precipitated with full-length TRRAP and domain 2 (TRRAP D2, amino acids 1898-2400) of TRRAP (Figure 2B). To explore whether TRRAP D2 can enhance the stability of NANOG, we examined NANOG protein stability in the absence or presence of TRRAP D2 after treatment with CHX. The protein stability of NANOG was enhanced by overexpressing TRRAP D2 (Figure 2C). To uncover the role of TRRAP D2 in NANOG ubiquitination, the ubiquitination levels of NANOG were measured by precipitating TAP-tagged NANOG and Western blot analysis of HA-Ub. Overexpression of the full-length and the D2 domain of TRRAP reduced the ubiquitination level of NANOG (Figure 2D).

### 2.3. TRRAP Interferes with the FBXW8-Mediated Ubiquitination of the NANOG Protein

FBXW8 has been reported as a ubiquitin E3 ligase for NANOG [14]. Therefore, we explored whether TRRAP affects the FBXW8-dependent ubiquitination of NANOG. FBXW8 overexpression significantly decreased the protein levels of NANOG in the cell lysates (Figure 3A). FBXW8 could be co-precipitated with NANOG, and FBXW8 overexpression led to an increase in the ubiquitination level of NANOG, suggesting that FBXW8 mediated the ubiquitination and proteasomal degradation of NANOG. Notably, the co-precipitation of FBXW8 with NANOG and the FBXW8-dependent degradation of NANOG was abrogated by TRRAP overexpression. Furthermore, the overexpression of not only full-length TRRAP but also TRRAP D2 inhibited the FBXW8-mediated ubiquitination and degradation of NANOG (Figure 3A,B). The overexpression of the full-length TRRAP or TRRAP D2 domain abrogated the interaction between the MYC-FBXW8 and TAP-NANOG (Figure 3A,B). These results suggest that TRRAP interferes with the binding of FBXW8 to NANOG, thereby abrogating the FBXW8-mediated ubiquitination and proteasomal degradation of NANOG.

### 2.4. Knockdown of TRRAP Attenuates the CSC-Like Properties of HCT-15 Spheroid Cells

To clarify the role of TRRAP in colorectal CSCs, we measured the expression levels of TRRAP and NANOG in adherent and three-dimensional spheroid cultures of HCT-15 colon cancer cells. The expression levels of TRRAP and NANOG were higher in the HCT-15 spheroids than in the adherent cells (Figure 4A). Moreover, the expression level of CD44, a colorectal CSC marker [21], was also higher in HCT-15 spheroids than in adherent cells. These results suggested that TRRAP and NANOG are overexpressed in colon CSCs.

To explore the role of TRRAP in regulating endogenous NANOG levels, TRRAP expression was silenced using TRRAP shRNA in HCT-15 spheroid cells. TRRAP knockdown reduced the protein levels of NANOG, CD44, and cyclin D1, in contrast to the increase in the protein level of P53, a well-known tumor suppressor gene in colorectal cancer [22], in response to TRRAP silencing (Figure 4B). It has been reported that NANOG [23] or CD44 [24] play key roles in developing resistance to cisplatin. Therefore, we examined the effects of TRRAP silencing on drug resistance in HCT-15 spheroids. Cisplatin treatment inhibited the viability of HCT-15 cells in a dose-dependent manner, and TRRAP silencing potentiated cisplatin-induced cell death (Figure 4C). Furthermore, TRRAP knockdown significantly decreased the spheroid-forming ability and proliferation of HCT-15 cells in three-dimensional suspension cultures (Figure 4D,E).

### 2.5. NANOG Overexpression Rescues the CSC-Like Characteristics of TRRAP-Silenced HCT-15 Spheroids

To explore whether the reduced expression of NANOG was responsible for the decreased drug resistance and proliferation of TRRAP-silenced HCT-15 cells, we overexpressed NANOG in TRRAP-silenced HCT-15 spheroid cells, followed by selection with G418. The overexpression of NANOG increased the expression levels of CD44 and cyclin D1; however, P53 expression was attenuated by NANOG overexpression (Figure 5A). Resistance to cisplatin markedly increased after NANOG overexpression compared to sh-TRRAP HCT-15 spheroid cells (Figure 5B). Additionally, it was observed that cell proliferation and spheroid-forming ability increased in response to NANOG overexpression (Figure 5C). These results suggest that TRRAP knockdown attenuates the CSC characteristics of HCT-15 spheroid cells via a NANOG-dependent mechanism.

### 2.6. Silencing of TRRAP Decreases In Vivo Tumor Growth

To investigate whether TRRAP is important for tumor growth, we measured the in vivo tumorigenicity of HCT-15 spheroid cells in a xenograft transplantation model. HCT-15 spheroid cells were infected with lentiviruses bearing control shRNA or TRRAP shRNA and cultured under selection with puromycin. The control and TRRAP-silenced HCT-15 spheroids were dissociated into single cells and then subcutaneously injected into the left and right sides of nude mice. Transplantation of TRRAP-silenced HCT-15 cells resulted in a significantly reduced tumor volume compared to control HCT-15 spheroid cells (Figure 6A). Moreover, transplantation of control HCT-15 cells increased tumor volume in a time-dependent manner, in contrast to the drastic inhibition of tumor growth observed in TRRAP-silenced HCT-15 cells (Figure 6B,C). These results indicated that TRRAP plays an important role in the in vivo tumor growth of HCT-15 spheroid cells by interfering with the FBXW8-mediated ubiquitination and proteolytic degradation of NANOG.

## 3. Discussion

An increased expression of NANOG is associated with poor survival in patients with cancer [25]. In the present study, we report that TRRAP is a novel NANOG-binding protein that regulates the protein stability of NANOG. Previously, we reported that the mRNA levels of *NANOG* were suppressed in TRRAP-silenced A2780 spheroid cells and that TRRAP overexpression increased it [20], suggesting that TRRAP has a role in the transcriptional regulation of NANOG. These results suggest that TRRAP regulates NANOG expression by regulating gene expression and protein stability. We also demonstrated that the D2 domain (amino acids 1898–2400) of TRRAP is responsible for its interaction with NANOG. The overexpression of the full-length and D2 domain of TRRAP interfered with the interaction of the E3 ubiquitin ligase FBXW8 with NANOG. These results support an interaction between TRRAP and NANOG in cells.

NANOG protein levels can be regulated by the ubiquitination-proteasomal pathway [12,26]. Ubiquitin-specific peptidase 21 (USP21) deubiquitylates NANOG but not OCT4 or SOX2. USP21 deubiquitylates the K48-type linkage in the ubiquitin chain of NANOG, thereby preventing its ubiquitin-dependent proteasomal degradation. Speckle-type POZ domain protein (SPOP), an adaptor of E3 ubiquitin ligase, is known to degrade NANOG. SPOP-mediated NANOG degradation is controlled by the AMPK-BRAF signaling axis through the phosphorylation of NANOG at Ser68, which blocks the interaction between SPOP and NANOG [27,28]. Cops2, a subunit of the COP9 signalosome, binds to the NANOG protein and prevents its proteasomal degradation [29]. In addition, the ERK1-dependent phosphorylation of NANOG induces the binding of FBXW8 with NANOG to reduce NANOG protein stability [14]. The present study demonstrated that TRRAP specifically stabilized NANOG protein, but not OCT4 or SOX2, regulating its ubiquitination by inhibiting its binding with the E3 ubiquitin ligase FBXW8. These results suggest that TRRAP is a NANOG-binding protein that regulates the protein stability of NANOG.

TRRAP expression is upregulated in spheroid cultures of A2780 ovarian cancer cells [20]. TRRAP knockdown significantly decreased cell proliferation and the number of A2780 spheroids. In addition, TRRAP knockdown induced cell cycle arrest and increased the percentage of apoptotic A2780 spheroids. Moreover, TRRAP knockdown significantly reduced in vivo tumor growth of A2780 cells. In the present study, we demonstrated increased TRRAP expression in HCT-15 colon adenocarcinoma spheroids. TRRAP knockdown inhibited tumor growth and NANOG expression in the HCT-15 xenograft model. Adding back NANOG abrogated the tumor growth suppressed by TRRAP silencing, suggesting the pivotal role of NANOG in TRRAP-mediated tumorigenesis. TRRAP knockdown decreased the expression of CD44, a CSC marker that regulates self-renewal, tumor initiation, and cancer metastasis [30]. In contrast, TRRAP silencing increased the expression of P53, which plays a key role in regulating cell cycle arrest or apoptosis in human cancer cells [31]. Furthermore, adding NANOG back to TRRAP-depleted cells restored the expression levels of CD44 and P53. These results suggest that TRRAP regulates cell proliferation and the maintenance of CSC characteristics by regulating CD44 and P53 expression in colon cancer cells in a NANOG-dependent manner.

The present study provides the first evidence that TRRAP plays an important role in the tumorigenic ability of colon cancer cells by regulating NANOG protein stability. We also demonstrated that NANOG is essential for maintaining the CSC-like properties of colon cancer spheroids. Taken together, these results suggest that TRRAP may be a key target gene for colon cancer treatment.

## 4. Materials and Methods

### 4.1. Materials

Fetal bovine serum, trypsin, and Hank’s balanced salt solution were purchased from Invitrogen (Carlsbad, CA, USA). Cell culture plates with ultralow attachment surfaces were purchased from Corning Life Sciences (Tewksbury, MA, USA). Neurobasal medium, FBS, B-27 Supplement, penicillin, and streptomycin were purchased from Life Technologies (Grand Island, NY, USA). TrypLE cell detachment solution was purchased from Thermo Fisher Scientific (Waltham, MA, USA). The HCT-15 cell line was obtained from the American Type Culture Collection. Anti-glyceraldehyde-3-phosphate dehydrogenase (GAPDH) antibodies were obtained from EMD Millipore (Billerica, MA, USA). M2 anti-FLAG antibodies were obtained from Sigma-Aldrich (Saint Louis, MO, USA). Antibodies against P53 (SC-6243) and OCT3/4 (SC-8628) were purchased from Santa Cruz Biotechnology Inc. (Dallas, TX, USA). Antibodies against cyclin D1 (ab134175), NANOG (ab109250), SOX2 (ab59776), and TRRAP (ab73546) were purchased from Abcam (Boston, MA, USA). Antibodies against CD44 (5640), MYC-Tag (2287S), and HA-Tag (3724S) were purchased from Cell Signaling Technology (Danvers, MA, USA).

### 4.2. Cell Culture

HEK293 cells were cultured in Dulbecco’s modified Eagle’s medium supplemented with 10% fetal bovine serum and antibiotics (100 U/mL penicillin and 100 μg/mL streptomycin) at 37 °C in a 5% CO_2_ atmosphere. Cell culture plates for adherent cells were purchased from Thermo Fisher Scientific. To create three-dimensional spheroid cultures, HCT-15 cells were seeded in an ultralow attachment dish at a density of 2 × 10^3^ cells/10 cm dish and cultured in a spheroid culture medium (neurobasal medium supplemented with B-27, 10 ng/mL basic fibroblast growth factor, 20 ng/mL epidermal growth factor, 2.5 μg/mL amphotericin B, 100 U/mL penicillin, and 100 μg/mL streptomycin). Fresh spheroid culture medium was added to the culture every 2 or 3 days. For the subculture of cancer spheroids, spheroids were dissociated into single cells via treatment with TrypLE^TM^ cell detachment solution, followed by filtering through a 40-μm cell strainer and plating at 1 × 10^4^ cells/mL in ultralow attachment dishes.

### 4.3. Co-Immunoprecipitation of TRRAP and NANOG

For the immunoprecipitation of TRRAP, a FLAG-TRRAP plasmid (CbS-Flag-TRRAP, Plasmid #32103, Addgene) and a TAP empty vector or TAP-NANOG vector were transfected into HEK293 cells using Lipofectamine PLUS reagent. Cell lysates were prepared via lysis in a cell lysis buffer (25 mM Tris-HCl [pH 8.0], 150 mM NaCl, 0.1% NP-40, 0.5 M EDTA, 0.1 mM β-mercaptoethanol, 1 mM PMSF, 5 mM protease inhibitor cocktail). An aliquot of the cell lysate (1 mg) was incubated with 50 µL (50% slurry) of an anti-FLAG M2 affinity gel (A2220, Sigma-Aldrich) for 2 h at 4 °C on a rotating wheel. The beads were then washed three times with the cell lysis buffer. Bound proteins were eluted by boiling in 2× SDS sample buffer and resolved on a 10% sodium dodecyl sulfate-polyacrylamide gel for Western blot analysis.

### 4.4. Measurement of Cell Death

To measure cisplatin-induced cell death, HCT-15 spheroid cells were dissociated into single cells and seeded in a 96-well culture plate at a density of 1 × 10^4^ cells/well. After culturing under various experimental conditions, the cells were incubated with 10 μL of EZ-Cytox (DoGenBio, Seoul, Republic of Korea,) solution for 1 h at 37 °C. Afterward, the solution was shaken for 1 min and the absorbance of the solution at 450 nm was determined using a micro-plate spectrophotometer (canton of Zürich, Switzerland TECAN). Cell counts were determined using a Trypan blue solution (Sigma-Aldrich).

### 4.5. Gene Transfection and Silencing

To overexpress TRRAP, OCT4, SOX2, and NANOG, HEK293 cells were transfected with the CbS-FLAG-TRRAP (Addgene, www.addgene.org, (accessed on 30 May 2014)), TAP-OCT4, TAP-SOX2, and TAP-NANOG vectors, respectively, using Lipofectamine and PLUS reagent. Two days after gene transfection, the mRNA and protein levels of the corresponding genes were measured via RT-PCR and Western blotting to confirm gene expression.

To generate lentiviruses bearing shRNAs targeting TRRAP, HEK293 cells were co-transfected with 6.67 μg of pLKO.1-puro lentiviral expression vectors bearing TRRAP-specific shRNA (GCCCTGTTCTTTCGCTTTGTA), 5 μg VSV-G envelope plasmid, and 3.33 μg Δ8.9 packaging plasmid using 15 μL Lipofectamine and Lipofectamine PLUS reagent. Culture supernatants containing lentivirus particles were harvested 48 h after transfection and concentrated using a Lenti-X Concentrator at 4 °C (Clontech Laboratories, Inc., Mountain View, CA, USA). For viral infection, HCT-15 cell spheroids were dissociated into single cells and infected via treatment with the concentrated viral supernatant in the presence of 5 μg/mL polybrene (Sigma-Aldrich) for 24 h. The lentivirus-infected cells were then selected by maintaining them in the spheroid culture medium supplemented with 0.5 μg/mL of puromycin. To overexpress NANOG in TRRAP-silenced HCT-15 cells, TRRAP shRNA lentivirus-infected HCT-15 cells were transfected with TAP empty or TAP-NANOG vectors using Lipofectamine and PLUS reagents, and then selected with 1 μg/mL of G418.

### 4.6. Measurement of Protein Stability and Ubiquitination

To explore the effect of TRRAP overexpression on the protein stability of NANOG, HEK293 cells were transfected with TAP-NANOG and FLAG-TRRAP expression vectors, followed by treatment with 50 μg/mL cycloheximide (CHX) to inhibit protein synthesis for the indicated times. The cell extracts were harvested, and the protein levels of NANOG were analyzed using Western blotting. To detect NANOG ubiquitination, the cells were co-transfected with plasmids bearing HA-ubiquitin and TAP-NANOG. Briefly, cell lysates were pretreated with 20 μM MG132 for 4 h and the level of ubiquitination of NANOG was measured via immunoprecipitation with streptavidin agarose beads (Agilent Technologies, Santa Clara, California, USA) and Western blot analysis using anti-HA antibodies.

### 4.7. Western Blotting Analysis

HEK293 and HCT-15 cells were lysed in a lysis buffer (20 mM Tris-HCl, 1 mM EGTA, 1 mM EDTA, 10 mM NaCl, 0.1 mM phenylmethyl sulfonyl fluoride, 1 mM Na_3_VO_4_, 30 mM sodium pyrophosphate, 25 mM β-glycerol phosphate, and 1% Triton X-100; pH 7.4). Lysates were resolved via sodium dodecyl sulfate-polyacrylamide gel electrophoresis, and the proteins were transferred to nitrocellulose membranes. The proteins were then stained with 0.1% Ponceau S solution (Sigma-Aldrich), blocked with 5% non-fat milk, and immunoblotted with antibodies overnight. Bound antibodies were visualized using the corresponding horseradish peroxidase-conjugated secondary antibodies using an enhanced chemiluminescence Western blotting system (GE Healthcare Life Sciences, Pittsburgh, PA, USA).

### 4.8. Reverse Transcription-Polymerase Chain Reaction (RT-PCR)

After the cells were harvested, total cellular RNA was extracted using TRIzol reagent (Sigma). mRNA levels were measured via RT-PCR using a Reverse Transcription cDNA kit (Nano helix, Daejeon, Republic of Korea). cDNA in 1 μL of the reaction mixture was amplified using a HelixAmp™ Ready-2×-Go PCR kit (Nano helix, www.nanohelix.co.kr, accessed on 3 April 2018) and 10 pmol of sense and antisense primers (sequences detailed below). The thermal profile was as follows: denaturation at 95 °C for 30 s, annealing at 51–55 °C for 30 s (depending on the primers used), and extension at 72 °C for 30 s. Each PCR reaction was conducted for 25–30 cycles. The PCR products were then analyzed using 1% agarose gel electrophoresis. The primer sequences used in this study are as follows: *GAPDH*, 5′-TCCATGACAA CTTTGGTATCG-3′, 5′-TGTAGCCAAATTCGTTGTCA-3′; *TRRAP*, 5′-ATGATGCAAGAAGTTAGTGAAAA-3′, 5′-GAAGATGTTCGTTGGTTGGTA-3′; *NANOG*, 5′-CGAGATAAACATGGCAATC AAAAT-3′, 5′-AATTCAGCAAGAAGCCTCTCCTT-3′; *OCT4*, 5′-GGGATATACACAGGCCGAT-3′, 5′-GATACTGGTTCGCTTTCTCT-3′; *SOX2*, 5′-GGGAAATGGGAGGGGTGCAAAAGAGG-3′, 5′-TTGCGTGAGTGTGGATGGGGATTGGTG-3′.

### 4.9. Xenograft Transplantation and In Vivo Monitoring

Mice were bred in a pathogen-free animal facility at Pusan National University under a rigorous monitoring system. Six-week-old female BALB/c-nu/nu mice were obtained from Orient Bio (Seongnam-si, Gyeonggi-do, Republic of Korea). In accordance with the guidelines of the Pusan National University pathogen-free animal facility, mice were acclimated for 1 week under a 12-h light/dark cycle, controlled temperature (25 ± 2 °C), and 50–60% relative humidity, with free access to water and standard rodent chow. The mice received a subcutaneous injection of HCT-15 cell spheroids (1 × 10^5^ cells/200 μL PBS) infected with the control or sh-TRRAP-bearing lentivirus. All injections were made in the right and left flanks of the mice. After 8 weeks, the mice were euthanized, the tumors were weighed, and the tumor volumes were measured. All animal experiments were conducted in accordance with the National Institutes of Health Guide for the Care and Use of Laboratory Animals and Institutional Animal Care and Use Committee (IACUC). The animal experimental protocol used in this study was approved by Pusan National University IACUC (No. PNU-2018-1816).

### 4.10. Statistical Analysis

Quantitative data are presented as the mean ± SD of representative experiments. The mean and SD values were calculated using Excel v2302 software. Significance was calculated using unpaired Student’s *t*-tests. For the statistical significance testing of multivariate datasets, one-way or two-way ANOVA with Scheffé’s test was used.

## Figures and Tables

**Figure 1 ijms-24-06260-f001:**
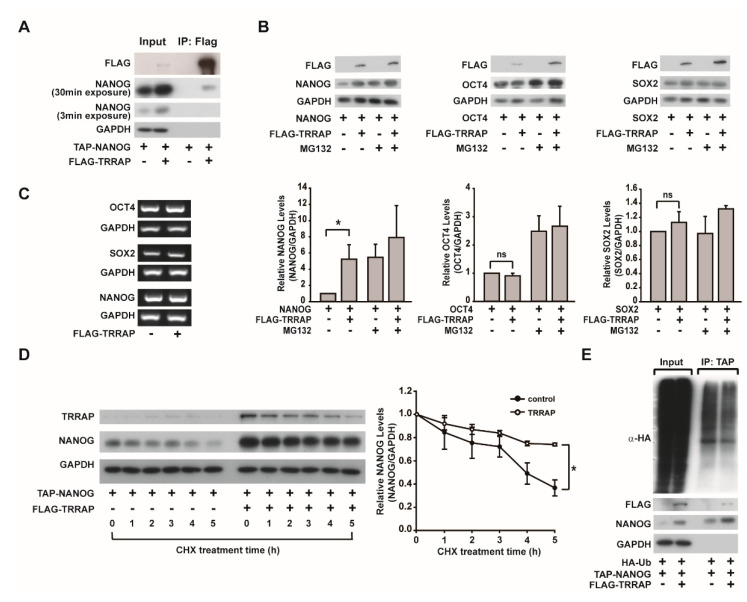
Identification of TRRAP as a NANOG-binding protein that affects NANOG stability. (**A**) Co-immunoprecipitation of NANOG and TRRAP in HEK293 cells. HEK293 cells were transfected with TAP-NANOG or FLAG-TRRAP, the cell lysates were incubated with streptavidin beads to precipitate TAP-NANOG, and the protein levels of GAPDH, NANOG, and TRRAP were analyzed via Western blotting analysis. (**B**) TRRAP-mediated increase in NANOG protein. TRRAP, OCT4, SOX2, and NANOG were overexpressed in HEK293 cells, which were then treated with MG132 for 4 h. Protein levels were analyzed via Western blot analysis. The protein levels of OCT4, SOX, and NANOG were quantified and normalized with that of GAPDH. Data are expressed as the mean ±  SD (*n*  =  3, lower panel). * indicates *p*  <  0.05 by Student’s *t*-test. ns stands for not significant. (**C**) The effects of TRRAP overexpression on the mRNA levels of *OCT4*, *SOX2*, and *NANOG*. HEK293 cells were transfected with TRRAP, OCT4, SOX2, and NANOG, and their mRNA levels were analyzed via RT-PCR. (**D**) The effects of TRRAP overexpression on the protein stability of NANOG. HEK293 cells were transfected with plasmids bearing TAP-NANOG and FLAG-TRRAP, treated with cycloheximide (CHX) for the indicated times, and the protein levels of TRRAP and NANOG were determined via Western blotting. The protein levels of NANOG were quantified and normalized with that of GAPDH (*n* = 3, right panel). * indicates *p*  <  0.05 by Student’s *t*-test. (**E**) The effects of TRRAP overexpression on the ubiquitination of NANOG. HEK293 cells were transfected with plasmids bearing TAP-NANOG, FLAG-TRRAP, and HA-Ub, and treated with 20 µM MG132 for 4 h, followed by the precipitation of TAP-NANOG with streptavidin beads. (Left panel) The protein and ubiquitination levels of NANOG were determined via Western blotting using anti-NANOG and anti-HA antibodies, respectively. (Right panel) Quantification of the relative levels of ubiquitinated NANOG per NANOG protein.

**Figure 2 ijms-24-06260-f002:**
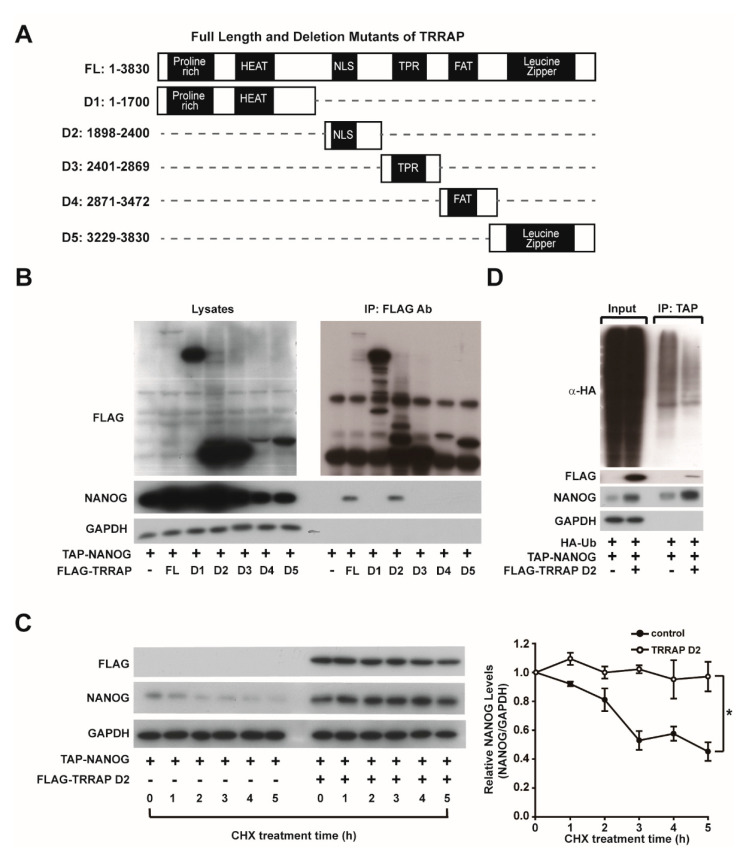
Identification of the TRRAP domain that regulates the ubiquitination of NANOG. (**A**) Schematic illustration of full-length and fragments of FLAG-tagged TRRAP. (**B**) Co-immunoprecipitation of FLAG-tagged TRRAP constructs and NANOG in HEK293 cells. HEK293 cells were transfected with TAP-NANOG, FLAG-TRRAP full-length, and its deletion mutants (D1–D5). FLAG-tagged TRRAP proteins were pulled down, and the protein levels of NANOG and TRRAP in the cell lysates and immunoprecipitants were determined via Western blotting. (**C**) Effects of the TRRAP D2 domain on the protein stability of NANOG. HEK293 cells were transfected with plasmids bearing TAP-NANOG and FLAG-TRRAP D2 domain, treated with cycloheximide (CHX) for the indicated times, and the protein levels of TRRAP and NANOG were determined via Western blotting. The protein levels of NANOG were then quantified and normalized with that of GAPDH (*n* = 3, right panel). * indicates *p*  <  0.05 by Student’s *t*-test. (**D**) Effect of the TRRAP D2 domain on the ubiquitination of NANOG. HEK293 cells were transfected with plasmids bearing TAP-NANOG, FLAG-TRRAP D2, and HA-Ub, and treated with 20 µM MG132 for 4 h. TAP-NANOG was then precipitated using streptavidin beads. The protein and ubiquitination levels of NANOG were determined via Western blotting with anti-NANOG and anti-HA antibodies, respectively.

**Figure 3 ijms-24-06260-f003:**
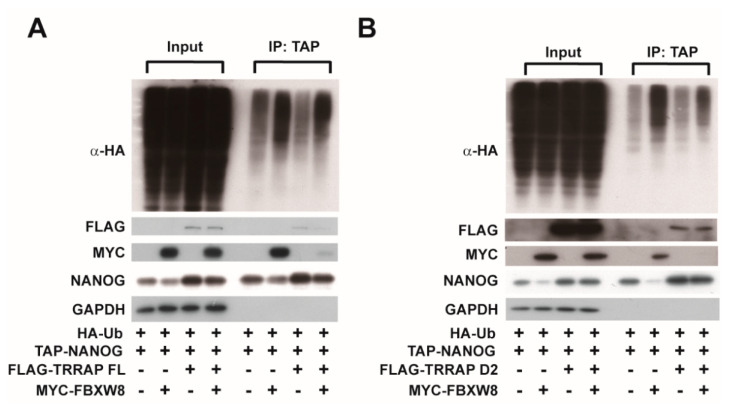
Effect of TRRAP overexpression on the FBXW8-mediated ubiquitination of the NANOG protein. HEK293 cells were transfected with plasmids expressing MYC-tagged FBXW8 and TAP-tagged NANOG along with FLAG-tagged full-length (FL) TRRAP (**A**) or its D2 domain only (**B**). The cells were then treated with 20 µM MG132 for 4 h, followed by the precipitation of TAP-NANOG using streptavidin beads. The ubiquitination levels of NANOG were determined via Western blotting using anti-HA antibodies. The protein levels of TRRAP, NANOG, FBXW8, and GAPDH in the cell lysates were analyzed via Western blotting.

**Figure 4 ijms-24-06260-f004:**
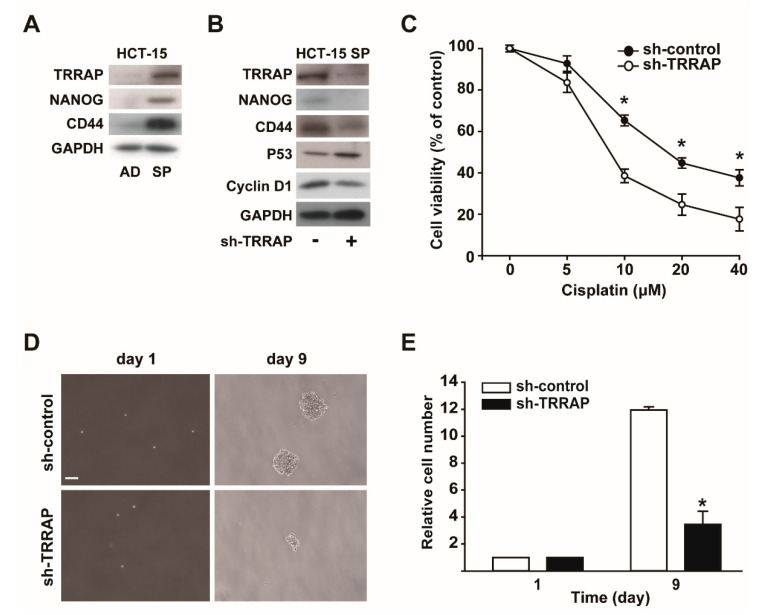
Effects of TRRAP silencing on the proliferation and drug resistance of HCT-15 spheroids. (**A**) The expressions of TRRAP and NANOG increased in spheroid cultures of HCT-15 cells. The expression of TRRAP, NANOG, CD44, and GAPDH were measured in adherent (AD) and spheroid (SP) cultures of HCT-15 colon cells. (**B**) Effects of the shRNA-mediated silencing of TRRAP on the expression levels of CSC-associated genes. HCT-15 spheroids (HCT-15 SP) were infected with or without lentiviruses bearing TRRAP shRNA, and the expression levels of TRRAP, NANOG, CD44, P53, cyclin D1, and GAPDH were measured via Western blot analysis. (**C**) Effects of TRRAP silencing on drug resistance in HCT-15 spheroids. HCT-15 spheroids were infected with lentiviruses bearing control shRNA (sh-control) or TRRAP shRNA (sh-TRRAP), treated with increasing concentrations of cisplatin for 24 h, and their cell viability was measured. * indicates *p*  <  0.05 by Student’s *t*-test. (**D**) Effects of TRRAP silencing on the spheroid-forming ability of HCT-15 cells. HCT-15 cells were infected with sh-control or sh-TRRAP lentiviruses, and bright field images of the spheroids were taken on days 1 and 9. Scale bar: 100 μm. (**E**) Effects of TRRAP silencing on the proliferation of HCT-15 spheroids. The cell numbers were counted on days 1 and 9 after TRRAP silencing. Results are presented as the mean ± SD. * indicates *p*  <  0.05 by Student’s *t*-test.

**Figure 5 ijms-24-06260-f005:**
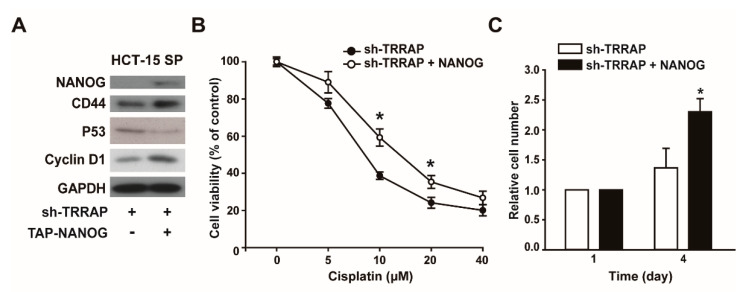
NANOG overexpression rescues the CSC-like characteristics of TRRAP-silenced HCT-15 spheroids. (**A**) Effects of NANOG overexpression on the expression of CD44, P53, and cyclin D1 in TRRAP-silenced HCT-15 spheroids. TRRAP-silenced HCT-15 spheroids were transfected with or without plasmids containing FLAG-NANOG, and the protein levels were determined via Western blotting. (**B**) Effects of NANOG overexpression on drug resistance in TRRAP-silenced HCT-15 spheroids. TRRAP-silenced HCT-15 spheroids were transfected with control or NANOG expression vectors, and treated with increasing concentrations of cisplatin for 24 h, followed by the measurement of cell viability. * indicates *p*  <  0.05 by Student’s *t*-test. (**C**) Effects of NANOG overexpression on the proliferation of TRRAP-silenced HCT-15 spheroids. The cell numbers were counted on days 1 and 4 after NANOG overexpression. Results are presented as the mean ± SD. * indicates *p*  <  0.05 by Student’s *t*-test.

**Figure 6 ijms-24-06260-f006:**
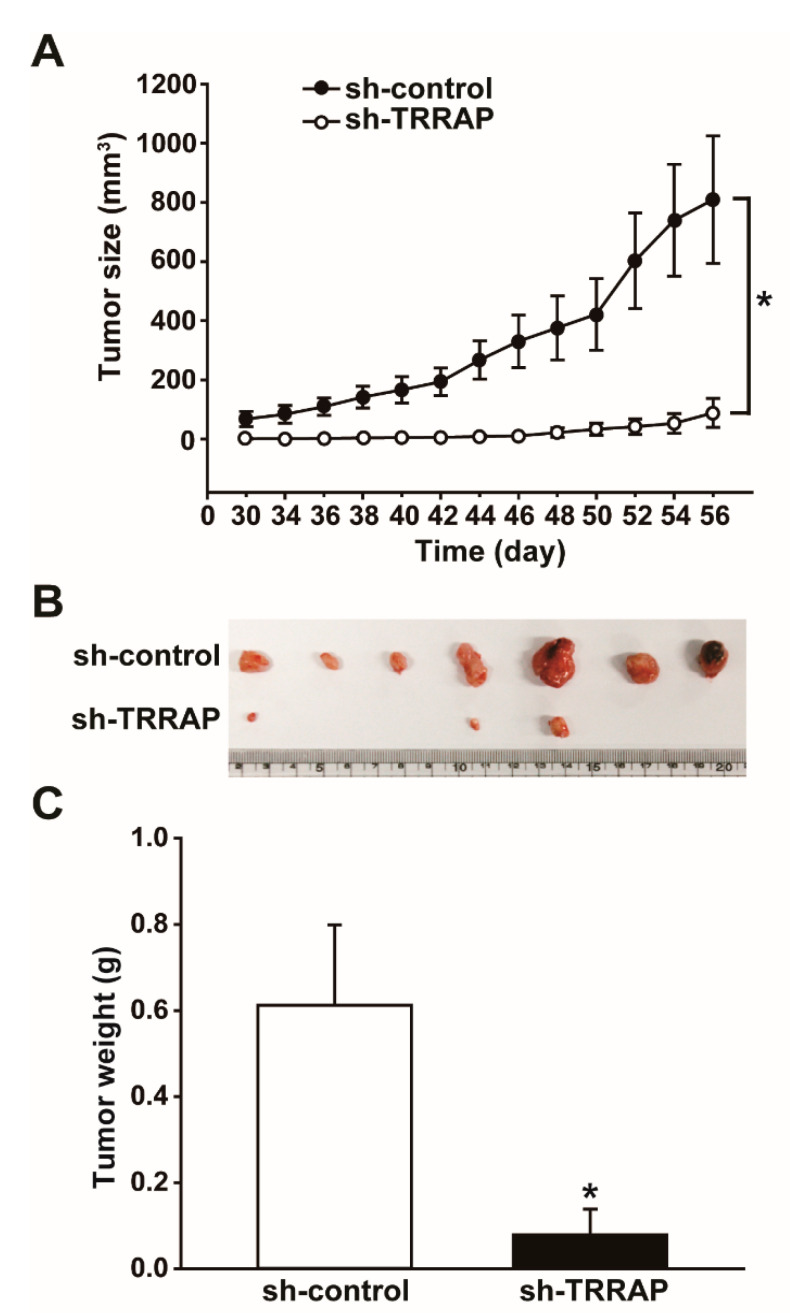
Effects of TRRAP silencing on the in vivo tumorigenic potential of HCT-15 spheroids. (**A**) Effects of TRRAP silencing on the in vivo growth of xenografts containing HCT-15 spheroids. HCT-15 spheroids were infected with lentiviruses expressing sh-control or sh-TRRAP and then transplanted into nude mice. Tumor volumes were measured from day 28 to 56 after injecting HCT-15 spheroids. * indicates *p*  <  0.05 by Student’s *t*-test. (**B**) Representative images of xenograft tumors isolated from the mice at 56 days after transplanting HCT-15 spheroids infected with sh-control or sh-TRRAP lentiviruses are shown. (**C**) Tumor weights were measured, and results are presented as the mean ± SD (*n* = 7). * indicates *p*  <  0.05 by Student’s *t*-test.

## Data Availability

The data reported in this article could be shared upon reasonable request to the corresponding author.

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
