# Peer review of "TRRAP Enhances Cancer Stem Cell Characteristics by Regulating NANOG Protein Stability in Colon Cancer Cells"

_ijms, 2023, doi:10.3390/ijms24076260_

Round 1

Reviewer 1 Report

1.    Did the author performed any experiment with purified proteins to check the interaction of TRRAP with NANOG?

2.    In Figure 1 and 2, the authors should mention how the error bars were calculated?

3.    The author should mention the exact time (short exposure and long exposure)

4.    The authors should repeat the western blot in figure 5A. The result is not conclusive

5. There are several grammatical errors in the manuscript.

Author Response

Response to Reviewer#1

Q1. Did the author performed any experiment with purified proteins to check the interaction of TRRAP with NANOG?

[Answer] Thank you for the reviewer’s valuable comments. According to the comment, we investigated whether TRRAP directly interacts with NANOG by in vitro pull-down experiment. TAP-NANOG and FLAG-TRRAP were overexpressed in HEK293 cells and the proteins were isolated by affinity purification with Streptavidin agarose and immunoprecipitation using anti-FLAG antibody, respectively. The purified TAP-NANOG and FLAG-TRRAP were incubated in vitro, followed by pull-down experiment for determination of direct binding of TAP-NANOG and FLAG-TRRAP. We found that TAP-NANOG directly interacts with FLAG-TRRAP in vitro (Supplementary Figure 1). We showed the result in the supplementary figure 1 and described it in Result section as follows.

Line 211: “To explore whether NANOG can directly interact with TRRAP, TAP-NANOG was purified via affinity purification with streptavidin agarose and eluted using biotin, followed by incubation with the immunoprecipitants of FLAG-TRRAP. TAP-NANOG could be pulled down by FLAG-TRRAP (Fig. S1), suggesting the direct interaction between TAP-NANOG and FLAG-TRRAP.”

Q2. In Figure 1 and 2, the authors should mention how the error bars were calculated?

[Answer] Thank you for the valuable comment. We calculated SD by using the equation (SD=σ/√n). According to the reviewer’s comments, we described about statistics in detail.

Line 196: “Quantitative data are presented as the mean ± SD of representative experiments. The mean and SD values were calculated using Excel software.”

Q3. The author should mention the exact time (short exposure and long exposure)

[Answer] According to the reviewer’s comments, we indicated the exposure times in the figure. Short exposure (3 min) and long exposure (30 min) were indicated in the figure 1.

Q4. The authors should repeat the western blot in figure 5A. The result is not conclusive.

[Answer] Thank you for the valuable comment. According to the reviewer’s comments, NANOG and CD44 were replaced with better images in Fig 5.

Q5. There are several grammatical errors in the manuscript.

[Answer] Thank you for the helpful comment. According to the reviewer’s comments, we carefully corrected the manuscript and all of grammatical errors by using a professional editing service (www.editage.co.kr).

Reviewer 2 Report

The manuscript entitled” TRRAP enhances cancer stem cell characteristics by regulating 2 NANOG protein stability in colon cancer cells” by Kyung Taek Kang et al. describes the role of TRRAP in enhancing stemness features, including stabilizing NANOG in colorectal cancer. The manuscript is well-designed, contains valuable data, and may aid translational insights into clinical practice. However, there are some minor concerns to be addressed:

1.       Line 61-73: I hope the authors could better explain the role of TRRAP, NANOG, and FBXW8 regarding their role in the transcriptional, translational, and post-translational regulation of NANPG. The current paragraph is not well-designed, even though it contains appropriate information.

2.       Line 216-220: It would be appreciated if authors explain why they carried on such an experiment and how they evaluated the specificity of TRRAP for NANOG upregulation. If they referred to protein stabilization, I seem that the term upregulation in this sentence was ambiguous.

3.       Line 298: How did you conclude that TRRAP interferes with the binding of FBXW8 to NANOG? You have only expressed that full-length TRRAP and TRRAP D2 inhibited the FBXW8-mediated ubiquitination and degradation of NANOG. Have you conducted other experiments to show competitive binding on TRRAP or FBXW8 to NANOG?

4.       Line 320-321: Doxorubicin is not the therapeutic choice for colorectal cancer treatment. It would be more valuable if authors used oxaliplatin or 5-FU in their experiments.

Best wishe,

17 February 2023

Author Response

Response to Reviewer#2

Q1. Line 61-73: I hope the authors could better explain the role of TRRAP, NANOG, and FBXW8 regarding their role in the transcriptional, translational, and post-translational regulation of NANPG. The current paragraph is not well-designed, even though it contains appropriate information.

[Answer] Thank you for the reviewer’s valuable comments. According to the comments, we revised paragraph to explain the information better as follows.

Line 48: “NANOG, a homeobox protein, is a transcription factor that contributes to embryonic stem cell pluripotency and self-renewal [6]. It has been reported that NANOG is a key regulator and transcription factor of CSCs and is upregulated in various human cancers [7]. NANOG has been reported to act as a tumor marker in patients with colorectal cancer and is associated with clinical and pathological features [8]. In human colorectal cancer, NANOG modulates stemness and tumorigenicity [9]. A high expression of NANOG in HCT116 cells causes increased colony formation and tumorigenicity by controlling the CSC population [10]. RNAi-mediated silencing of NANOG expression suppressed proliferation and increased apoptosis in the EpCAM+/CD44+ CSC population of HCT116 colorectal cancer cells [11]. In addition to transcriptional regulation, NANOG protein levels are regulated by ubiquitination and proteasomal degradation [12], [13]. FBXW8 is a well-known ubiquitin ligase that regulates NANOG protein stability by mediating its ubiquitination [14]. However, the molecular mechanisms underlying the regulation of NANOG protein stability have not yet been elucidated.

TRRAP is a large adaptor protein with homology similar to PIKK kinases [15]. TRRAP promotes histone acetylation and chromatin remodeling and regulates gene expression and embryonic development [16]. TRRAP depletion has been shown to decrease the expression of the stemness-associated genes OCT4, SOX2, and NANOG and increase the expression of differentiation markers from the germ layers [16]. TRRAP expression levels have been reported to be significantly upregulated in breast cancer [17], while the TRRAP knockdown reduced the CSC-like properties of glioma [18,19]. TRRAP overexpression also increased the mRNA levels of NANOG, while TRRAP knockdown reduced tumor growth in a murine ovarian cancer xenograft model [20]. However, although TRRAP and NANOG play an important role in the tumorigenicity of CSCs, the role of TRRAP in regulating NANOG protein stability remains uncertain.”

Q2. Line 216-220: It would be appreciated if authors explain why they carried on such an experiment and how they evaluated the specificity of TRRAP for NANOG upregulation. If they referred to protein stabilization, I seem that the term upregulation in this sentence was ambiguous.

[Answer] According to the reviewer’s comments, we added explanation and changed to more appropriate sentence.

Line 217: “To explore whether NANOG protein levels are specifically increased by TRRAP overexpression, we overexpressed OCT4, SOX2, or NANOG along with TRRAP. In contrast to the TRRAP-dependent increase in NANOG protein levels, there was no significant change in OCT4 and SOX2 protein levels (Fig. 1B). Treatment of HEK293 cells with the proteasome inhibitor MG132 to block proteasomal degradation of proteins increased the protein levels of NANOG and OCT4. However, the mRNA levels of OCT4, SOX2, and NANOG were not significantly affected by TRRAP overexpression (Fig. 1C). These results suggest that TRRAP specifically increased the protein level of NANOG, but not OCT4 or SOX2, suggesting the TRRAP-dependent regulation of NANOG protein stability.”

Q3. Line 298: How did you conclude that TRRAP interferes with the binding of FBXW8 to NANOG? You have only expressed that full-length TRRAP and TRRAP D2 inhibited the FBXW8-mediated ubiquitination and degradation of NANOG. Have you conducted other experiments to show competitive binding on TRRAP or FBXW8 to NANOG?

[Answer] Thank you for the valuable comment. From the co-immunoprecipitation experiment, we found that overexpression of the full length TRRAP or the TRRAP D2 domain abrogated the binding of FBXW8 and NANOG. We showed these results in Figure 3 and described it in Result section as follows.

Line 300: “The overexpression of the full length TRRAP or TRRAP D2 domain abrogated the interaction between the MYC-FBXW8 and TAP-NANOG (Fig. 3A and 3B).”

Q4. Line 320-321: Doxorubicin is not the therapeutic choice for colorectal cancer treatment. It would be more valuable if authors used oxaliplatin or 5-FU in their experiments.

[Answer] According to the reviewer’s comments, we examined the role of TRRAP and NANOG on drug resistance of HCT-15 spheroids against not only doxorubicin but also cisplatin. We replaced the doxorubicin data with the cisplatin data in Figure 4C and 5B.

Round 2

Reviewer 1 Report

The result in Figure S1 is confusing.

1. The author claims that TAP-NANOG directly interacts with FLAG-TRRAP in vitro (Supplementary Figure 1). However in the control Flag lane also, we see signal for NANOG. So it indicates that the interaction is nonspecific or the experiment has not been performed correctly.

Author Response

Thank you for your valuable comment. To confirm the direct interaction of TAP-NANOG with FLAG-TRRAP, we incubated purified TAP-NANOG protein with FLAG-TRRAP bound to anti-FLAG antibody immobilized agarose. As a negative control, we incubated TAP-NANOG protein with anti-FLAG antibody immobilized agarose. As pointed out by the reviewer, a small amount of TAP-NANOG could bind to anti-FLAG antibody immobilized agarose, suggesting a non-specific interaction between TAP-NANOG and control beads. In spite of the non-specific binding between TAP-NANOG and control beads, the band intensity of TAP-NANOG bound to FLAG-TRRAP beads was much greater than the band intensity bound to control beads. These results suggest a direct interaction between TAP-NANOG and FLAG-TRRAP.

To clearly demonstrate the result, we replaced the supplementary figure 1C with shorter exposed western blot data.

Round 3

Reviewer 1 Report

I have no further comments